# A Novel Noninvasive Technique for Intracranial Pressure Waveform Monitoring in Critical Care

**DOI:** 10.3390/jpm11121302

**Published:** 2021-12-05

**Authors:** Sérgio Brasil, Davi Jorge Fontoura Solla, Ricardo de Carvalho Nogueira, Manoel Jacobsen Teixeira, Luiz Marcelo Sá Malbouisson, Wellingson da Silva Paiva

**Affiliations:** 1Department of Neurology, School of Medicine, University of São Paulo, São Paulo 01246, Brazil; davisolla@gmail.com (D.J.F.S.); rcnogueira28@gmail.com (R.d.C.N.); manoeljacobsen@gmail.com (M.J.T.); wellingsonpaiva@yahoo.com.br (W.d.S.P.); 2Department of Intensive Care, School of Medicine, University of São Paulo, São Paulo 01246, Brazil; luiz.malbouisson@hc.fm.usp.br

**Keywords:** intracranial compliance, intracranial pressure, intracranial hypertension, acute brain injury

## Abstract

Background: We validated a new noninvasive tool (B4C) to assess intracranial pressure waveform (ICPW) morphology in a set of neurocritical patients, correlating the data with ICPW obtained from invasive catheter monitoring. Materials and Methods: Patients undergoing invasive intracranial pressure (ICP) monitoring were consecutively evaluated using the B4C sensor. Ultrasound-guided manual internal jugular vein (IJV) compression was performed to elevate ICP from the baseline. ICP values, amplitudes, and time intervals (P2/P1 ratio and time-to-peak [TTP]) between the ICP and B4C waveform peaks were analyzed. Results: Among 41 patients, the main causes for ICP monitoring included traumatic brain injury, subarachnoid hemorrhage, and stroke. Bland–Altman’s plot indicated agreement between the ICPW parameters obtained using both techniques. The strongest Pearson’s correlation for P2/P1 and TTP was observed among patients with no cranial damage (r = 0.72 and 0.85, respectively) to the detriment of those who have undergone craniotomies or craniectomies. P2/P1 values of 1 were equivalent between the two techniques (area under the receiver operator curve [AUROC], 0.9) whereas B4C cut-off 1.2 was predictive of intracranial hypertension (AUROC 0.9, *p* < 000.1 for ICP > 20 mmHg). Conclusion: B4C provided biometric amplitude ratios correlated with ICPW variation morphology and is useful for noninvasive critical care monitoring.

## 1. Introduction

Continuous invasive intracranial pressure (ICP) monitoring is crucial for the management of intracranial hypertension (ICH) in the intensive care setting [1,2]. Conventional ICP monitoring methods include trepanation and insertion of a catheter through the skull for quantitative ICP measurements [3,4,5,6]. However, this procedure exposes the patients to several risks that limit its applicability [7,8,9]. Moreover, ICP monitoring is not always available for patients in need in many places worldwide [2]. These disadvantages indicate the need for a less invasive method of ICP monitoring [10,11,12,13].

Noninvasive surrogates for ICP monitoring have been proposed, especially bedside techniques as pupillometry and ultrasound methods such as transcranial Doppler and optic nerve sheath measure have been applied for the assessment of intracranial compliance (ICC) [14,15,16,17]. ICC expresses the relationship between intracranial components’ volumes (brain, cerebrospinal fluid, and blood) and the loss of hemostasis between them is the cause of substantial increases in ICP [18,19]. Moreover, ICC seems to be a more accurate indicator of loss of intracranial hemostasis than ICP mean values itself [20].

Another method to observe ICC impairment is the intracranial pressure waveform (ICPW) [15] by the alteration of the natural relation of its peaks; P1 (arterial pulsation), P2 (cerebral venous flow, which is secondary to cyclic fluctuations of arterial blood volume, Figure 1), and P3 (aortic valve closure) [21].

Exploration of ICPW has been a field of development and ICC understanding restricted to invasive techniques, nevertheless, a mechanical sensor (B4C) placed in contact with cranial skin has been recently developed with the capacity for acquiring ICPW. The system detects beat-by-beat micrometric cranial deformations. Experimental models in vitro [6] and with saline infusion on the spinal space in rats and pigs have demonstrated a linear correlation between ICP and skull deformation [3,22]. However, the application of this new method in clinical practice is yet to be determined [3,4,5,6,23,24]. Therefore, the primary endpoint of this study was the comparison of ICPW parameters obtained through invasive catheter monitoring with the waveforms obtained by the B4C sensor, in a set of neurocritical patients. The secondary endpoint was to verify whether loss of skull bone integrity could have hindrances to ICPW assessment.

## 2. Methods

This single-center observational prospective study involving six intensive care units (ICUs) of the Hospital das Clínicas, São Paulo University, Brazil had received ethics committee approval and has been ongoing since 2017. Consecutive patients are being recruited; however, due to the 2019 coronavirus disease (COVID-19) pandemic, patient recruitment has been temporarily interrupted. The clinical trial study protocol was approved by the local Ethics Committee in April 2017 and was registered under number NCT03144219 (available at http://clinicaltrials.gov, 4 December 2021). The B4C system was included in the same research protocol in August 2019. All methods were performed following the relevant guidelines and regulations, and informed consent was obtained from all legally authorized representatives/next of kin instead of the patients because of illness severity.

### 2.1. Study Design

All patients included in the study had suffered either traumatic or non-traumatic acute brain injury with the need for ventilatory support and are under invasive ICP monitoring according to the neurosurgical guidelines adopted by our institution. Data collection consisted of a single 10-min session for each patient, with simultaneous recording of invasive arterial blood pressure, ICP, B4C, ECG, and oxygen saturation in spontaneous variations. At minute 7, ultrasound-guided manual internal jugular vein (IJV) compression was performed for 60 s. With the patient’s head positioned at 30°, intracranial blood drainage becomes almost completely shifted to the IJVs, and compressing these veins is effective in increasing intracranial volume [25,26]. These short sessions were performed to avoid the occurrence of substantial changes in systemic parameters during recording and to observe the impact of slight intracranial volume damming on ICP values and ICPW parameters. This maneuver consequently generated a 60-s plateau wave in both techniques’ registers (Figure 2). Data analyses were based on a comparison of baseline ICPW parameters with the same parameters during IJV compression. As these variables were continuously recorded during the procedure, an average of 700 pulses for analysis was expected from each patient. This study protocol also followed the Standards for Reporting of Diagnostic Accuracy Studies (STARD) (Appendix A).

### 2.2. Participants

The inclusion criteria consisted of neurocritical patients of any cause or sex, >18 years, and who underwent ICP monitoring displaying both ICP numeric values and waveforms until the 5th day of catheter insertion. Patients were included according to their admission to prevent selection by convenience. Data obtained using the B4C sensor were not used for clinical management. We excluded patients presenting fixed mydriatic or middle-sized pupils for more than 2 h after ventilatory and hemodynamic stabilization. For analytic purposes, the patients were grouped with reference to intact skull bone (group A), large skull fractures or craniotomies (group B), and craniectomies (group C).

### 2.3. Clinical Variables

The clinical variables collected were age in years (continuous variable), diagnostic variables, the Marshall tomographic score in the case of TBI, the modified Fisher tomographic score in case of SAH, arterial blood pressure, axillary temperature, heart and respiratory rates, oxygen saturation, and sedatives administered.

### 2.4. Invasive ICP Monitoring (Gold Standard)

We used an intraventricular measurement system as the standard method. The ICP Neurovent monitoring system (Raumedic^®^, Munchberg, Germany) consisted of a pressure probe for ventricular use. This system can be attached to any monitor using a small zero-point-specific simulator for patient monitoring. Changes in the monitor during measurement do not result in a loss of calibration. The function was based on an electronic chip at its end. This membrane protrudes from the degree of pressure to which it is exposed. The pressure was measured by determining the membrane deformity using the piezoelectric system. The required measurement accuracy and independence of the inlet pressure variations were ensured by an integrated measuring bridge on the chip.

### 2.5. Noninvasive Intracranial Compliance Monitoring (B4C)

The B4C (B4C; Brain4care Corp., São Carlos, Brazil) sensor consisted of support for a sensor bar that detects local cranial bone deformations using specific sensors [27]. The detection of these deformations was obtained using a cantilever bar modeled through finite element calculations. Voltage meters were attached to this bar for deformation detection. Noninvasive contact with the skull was obtained by adequate direct pressure onto the scalp using a pin. The system was positioned in the frontotemporal region, approximately 3 cm over the first third of the orbitomeatal line. Consequently, the main branches of the temporal superficial artery and the temporal muscle were avoided, and sensor contact was provided through an area of thin skin and skull bone, whereas slight pressure was applied to the adjustable band until an optimal signal was detected.

Variations in ICP cause micrometric deformations in the cranial bone, which are detected by the sensor bar with a sensitivity to register cranial movements of <0.2 micrometers. The device filtered, amplified, and scanned the sensor signal. It also sent the data to a mobile device. The method was completely non-invasive and painless. In addition, it did not interfere with or had been jammed by routine monitoring. The obtained waveform was equivalent to the ICP waveform obtained through invasive techniques, such as intraparenchymal probes or external ventricular derivation, hence the ICPW was reproduced [3].

The B4C analytics system verified all data collected by the sensor, that is, ICP pulse wave morphology parameters such as the P2/P1 ratio, time-to-peak (TTP) interval, and pulse amplitudes [28]. For this study, all calculations were performed from the average of the pulses within each minute of monitoring after excluding possible artifacts. These averages were used to calculate the amplitudes of the two main ICPW peaks, P2 and P1. The P2/P1 ratio was calculated by dividing the amplitudes of the two peaks. All B4C sensor data obtained is processed by an algorithm previously created from the synchronization of B4C signals with arterial blood pressure obtained from more than a hundred thousand heartbeats [27]. As the cardiac cycle may be overlapped with the respiratory cycle [29], this automated system indicates where P2 should be depicted on the waveform, instead of where it appears on the spectrum [29] (Figure 3). A comprehensive explanation of B4C system functioning has been published [27]. The same algorithm was applied for the analysis of invasive ICPW features in another recently published study [30].

### 2.6. Sample

For the preliminary description of the findings for this new technique, the desired sample consisted of 40 consecutive subjects [31]. Nevertheless, data collection is still ongoing because of the high prevalence of ICH among neurocritical patients.

### 2.7. Statistical Analyses

For descriptive purposes, categorical variables were presented through relative and absolute frequencies and were compared using the chi-square or Fisher’s exact test, as appropriate. Continuous variable distributions were deemed normal, as assessed by skewness, kurtosis, and graphical methods. There were no missing data for intracranial monitoring parameters. The statistical analyses consisted of the Bland–Altman correspondence plot and linear fit using the QtiPlot v5.14.2 software (available at www.qtiplot.com). Additionally, a linear correlation was presented using R. The ROC curve analysis was performed using the Johns Hopkins University tool (available at www.jrocfit.org). Pearson’s correlation was calculated using 95% confidence intervals obtained via bootstrapping.

## 3. Results

### 3.1. Sample Features

The presented results refer to 41 consecutive patients admitted to our institution between August 2019 and May 2020 who have undergone ICP monitoring. The overall clinical features are presented in Table 1. No reports of adverse events of any nature have been reported.

### 3.2. Correlation between ICP and B4C

The data were pooled from one 10-min session for each of the 41 patients, resulting in a total sample size of 29.458 cardiac pulses. The Bland–Altman plots for the P2/P1 ratio and TTP for both baseline and with respect to IJV (Figure 4). Thresholds were defined as 1 for P2/P1 ratio and 0.2 for TTP within a 95% limit of agreement.

Patients with preserved cranial integrity (group A) exhibited the best linear correlations for both P2/P1 ratio and TTP (r = 0.72 and 0.85, respectively; Figure 5). The dispersion plots, in consideration of the IJV compression intervention, depicted similar behavior between the invasive and noninvasive morphology variations, with elevations of P2/P1 ratios for groups A and B. Table 2 shows the mean ICP value elevation, **Δ**, and the percentage of P2/P1 ratio variations with the intervention for intracranial volume damming applied. For patients in group C, the IJV compression led to a drop in the P2/P1 ratio, as verified by both techniques (Figure 6).

The IJV compression was effective in promoting intracranial volume damming and consequently ICP elevation, as 36% of patients overpassed 20 mmHg ICP cut-off during monitoring; therefore, the area under the receiver operator curve for different P2/P1 ratio cut-offs for ICP values >20 mmHg was calculated (Figure 7).

For our full sample, power to discriminate ICH (ICP > 20 mmHg) with B4C P2/P1 ratio threshold of ≥1.1 was AUC 0.77 (95% CI 0.62–0.92, *p* < 0.001, sensitivity 0.88, specificity 0.60). This accuracy appeared to be superior for those with intact skulls (AUC 0.90, P2/P1 ratio threshold ≥ 1.2, *p* < 0.001) compared to craniotomy/fracture (AUC 0.78, P2/P1 ratio threshold ≥ 1.1) or craniectomy (AUC 0.70, P2/P1 ratio threshold ≥ 1.1), but the comparisons are hindered by the small sample size and limited power, and the differences didn’t reach statistical significance. The invasive ICP P2/P1 ratio accuracy to discriminate ICH (ICP > 20 mmHg) was similar to B4C’s, as well as the optimal cutoffs.

## 4. Discussion

### 4.1. Main Findings

The results of this study demonstrated a statistically significant correlation between the ICPW morphology parameters of the gold standard invasive ICP monitoring and the B4C system. The slight variation in intracranial volume conducted in this study was effective in promoting changes in the ICP waveform parameters that were similarly observed from the inside and outside of the skull. The Bland–Altman analysis showed agreement between invasive and noninvasive ICPW features, with 4–7% outliers in this sample, although the majority of the measurements were scattered near to the no difference line. Nevertheless, invasive and noninvasive waveform morphologies were better correlated among themselves than with the ICP values. This suggests that despite having good correspondence, ICC and ICP may have different paces, reinforcing the need to look for ICP beyond its numbers. The noninvasive system has been demonstrated to be suitable for observing the hemostasis of intracranial component volumes and could be a promising surrogate for ICC monitoring with real-time resolution; absolute ICP (mmHg) values are not disclosed, but a reliable P2/P1 ratio and TTP could be obtained with power for predicting ICH when the ratio is >1.2 for patients with cranial integrity (AUC 0.9, *p* < 0.001). The same threshold may not be applicable to craniectomized or operated patients, although the system may remain useful in these situations because of the possibility of continuous monitoring without adding risks.

A perfect correlation between B4C and ICP waveform parameters could not be expected, since the measurement locations differ—the invasive one is mostly placed into the ventricle, whereas the B4C is placed outside the skull/scalp. The physical properties of the new technology and the intracranial catheters also differ (mmHg against skull expansion in µm). Moreover, our sample was heterogeneous, including patients with TBI, SAH, and stroke, and our sample also had a high percentage of patients who have undergone surgery. Nevertheless, our findings are still significant, especially because of the verification of similar behavior for the ICP curves obtained using both techniques, whether for intact or damaged skulls or even for patients who have undergone craniectomies. 

Our sample consisted of many patients without skull integrity (71%). Our lowest Pearson’s correlation between techniques for both P2/P1 ratio and TTP was observed for group B, patients presenting large fractures or post-surgical cranial manipulation, probably for the higher heterogeneity among the 20 patients in this group, whereas for groups A and C cranial condition was either integrated or largely opened respectively, leading to satisfactory correlations for these groups. Patients who have undergone craniectomies generally have a lower influence of ICP elevation on ICPW morphology [30], which is a phenomenon that was also observed in the present study using both techniques, indicating the need for the interaction between the intracranial content and the skull bone for proper ICC maintenance. Moreover, the possibility of either ICC impairment coexisting with ICP within the normal range or proper ICC coexisting with altered ICP values has been described. Therefore, continuous leashing of ICC to the mean ICP values is controversial [20,32]. Thus, the ICPW-derived indices continuously provided by this technique could be of value. The P2/P1 ratio has been proposed as a marker of ICC [33], and the enlargement of intracranial pulse shapes, the peak amplitudes, and the time interval between these peaks when ICH is present have also been previously described using data obtained from invasive systems [34,35,36].

The first clinical study that applied this new technology successfully correlated the P2/P1 ratio with ICP in children with hydrocephalus [24], whereas another study assessed cerebrovascular disorders using transcranial Doppler and B4C devices in patients with severe COVID-19. In this study, a progressive score from 5 to 20 was created by combining the results of both techniques, indicating that the worse the cerebral hemodynamics and B4C P2/P1 ratio alterations, the higher the probability of either the impossibility of mechanical ventilation weaning or even death until seven days after examinations (*p* < 0.00001) [37], with a B4C P2/P1 ratio alteration, observed more frequently among patients who are obese (*p* = 0.029) [38]. Another study has observed an elevation in the P2/P1 ratio and TTP features extracted from B4C waveforms in patients with end-stage renal disease before and after hemodialysis sessions (*p* < 0.01), suggesting that ICC is disturbed in these patients [39]. 

### 4.2. Limitations of the Study

This study was designed to perform direct observation and correlation between the parameters of both techniques in a single short session; hence, no outcome analysis for the included patients had been suitable. Intracranial compliance is a concept derived from volume and pressure variation; hence, as in the present study, the volume variation was based on the individuals’ characteristics rather than being controlled, and we could not determine the exact behavior of this system in correspondence with volume variations at this time. The main limitation of the system is the need for patient cooperation for those who are awake since, in cases of agitation, the sensor will not maintain ideal contact with the cranial skin. In cases of decompressive craniectomy, the system can still be used because there is enough rigid structure to comprehend the band, with the sensor placed on the integrated side of the skull.

Other limitations for consideration include the system’s unsuitability for preterm neonates (cranial vault < 47 cm). Furthermore, as the system maintains contact with the skin, it should be relocated hourly to avoid scalp erythema. Finally, while the B4C system presents analytical capabilities to provide additional ICC waveform information to the clinician, clinicians should always use their professional judgment to determine additional interventions necessary in the management of their patients.

### 4.3. B4C System Attributes

The utilization of this noninvasive system offers no additional risk to patients. The reports produced may be tracked anywhere, permitting the physician to monitor either one or several patients remotely. The application and data collection may also be performed by a trained technician. For validation purposes, the present study was a short session designed to avoid including confounding factors, such as ABP large variations per example, and focusing on an exclusive ICP variation fashion. The correlations then were made beat-by-beat between ICP and B4C pulses. However, the system is suitable for continuous long-term monitoring or multiple serials monitoring sessions according to physicians’ judgments, either aiding screening for invasive ICP monitoring per example, but also allowing the observation of ABP, sedation and ventilation changes. Furthermore, the waveform morphologies could reflect the residual compensatory capacity of the brain (Lundberg waves) [35], for subjects with impaired cerebrovascular autoregulation and the lower ability for vasodilation and contraction [40]. The aforementioned gives the ICP waveform a remarkable role in the care of critical patients, since changes in the ICP wave shape may be informative for an incoming or established alteration of the intracranial system [21,41]. Further studies will estimate the impact of this information on this population’s outcomes.

### 4.4. Considerations for the Future

Despite strong recommendations, class I evidence for each modality of monitoring in the neuro ICU remains lacking [42,43,44], except for ICP monitoring in cases of imminence for brain herniation [44]. The hindrances to reaching high levels of evidence are more related to the difficulties in designing randomized controlled trials in the field than to the techniques’ particularities. In the case of ICP, randomizing patients exclusively for studying purposes remain controversial [45]. As this new system for ICC monitoring does not have additional risks, it represents an option for the development of trials in critical care, since ICC impairment are not exclusive to CNS primary diseases, but could also be observed in situations of severe acute respiratory syndrome; cardiac, hepatic [46], or kidney failure [47]; anesthetics; and extracorporeal membrane oxygenation [48], among others.

## 5. Conclusions

Our study demonstrated that micrometric beat-by-beat cranial pulses provided by the B4C device are comparable to the invasive intracranial pressure pulse morphology. The ICP waveform obtained noninvasively through this system may widen the applicability and understanding of ICP in less explored clinical fields.

## Figures and Tables

**Figure 1 jpm-11-01302-f001:**
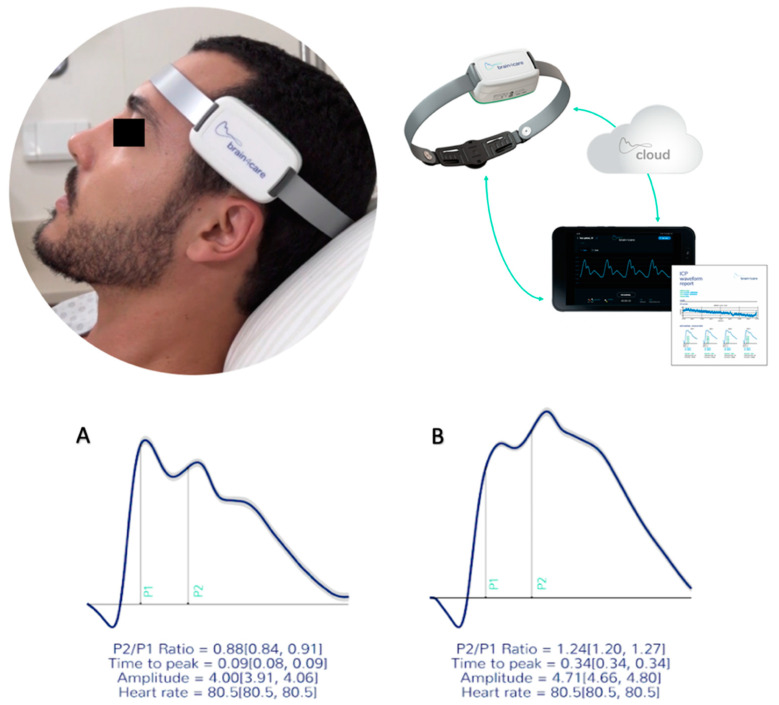
The B4C device in use; real time monitoring is displayed in a common portable device. All data collected is immediately processed on the cloud resulting in the qualitative and quantitative report of intracranial pressure waveforms obtained noninvasively. Waveforms obtained depicting P2/P1 ratio and time-to-peak under normal standards (**A**) and altered (**B**).

**Figure 2 jpm-11-01302-f002:**
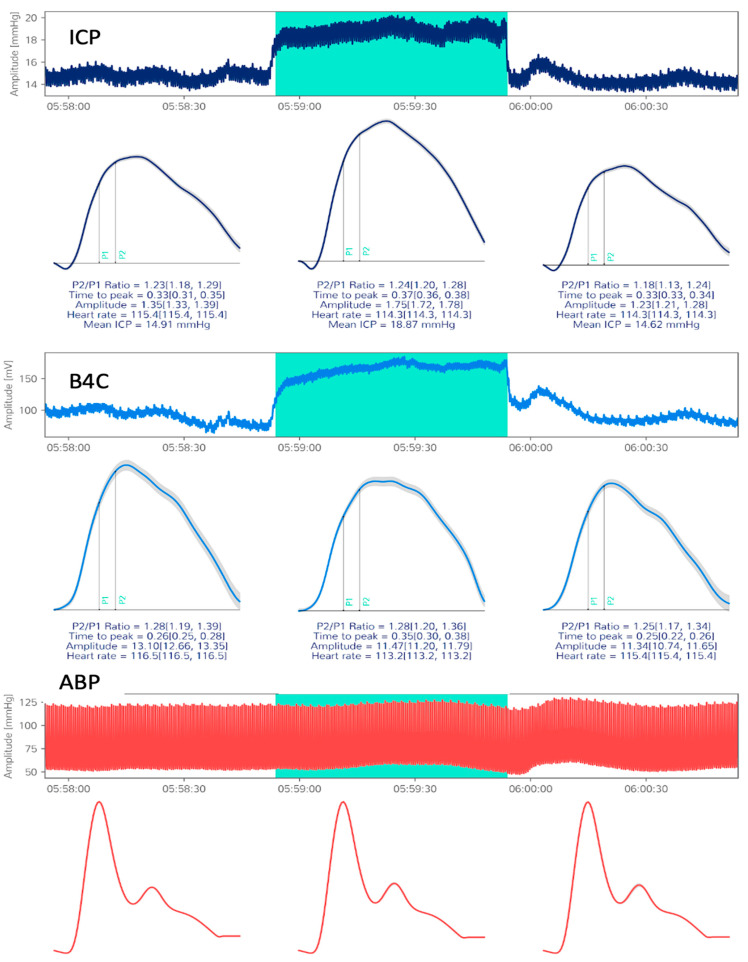
Sample of patient 36, craniectomized after severe TBI. Compacted ICP (mmHg), B4C (µm) and ABP recordings, with the mean-pulse morphologies depicted of the period pre-compression (**left**), during compression (**middle**, **green zone**) and posterior to IJVs compression (**right**). The automated algorithm based on the ABP cycle localized P1 and P2 on ICP and B4C pulses, performing P2/P1 ratio and time-to-peak calculations. ICP: intracranial pressure, B4C: brain4care device, ABP: arterial blood pressure, IJVs: internal jugular veins, TBI: traumatic brain injury.

**Figure 3 jpm-11-01302-f003:**
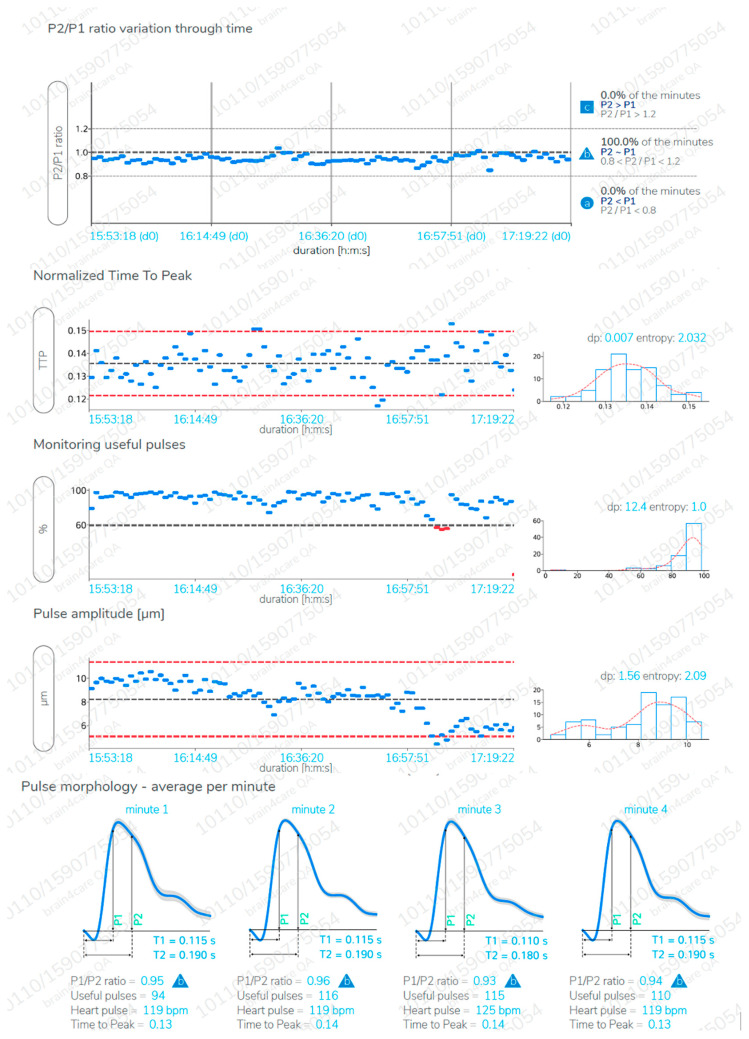
A B4C monitoring report sample of more than an hour length. Real time report is obtained without monitoring interruption as needed. P2/P1 ratio, time to peak, pulse amplitudes and useful pulses (pulses that were successfully recognized) are disclosed for the entire monitoring (each blue point is the mean value for a minute) and a mean waveform with confidence interval (gray zone along the slope) of each monitored minute is also displayed (bottom).

**Figure 4 jpm-11-01302-f004:**
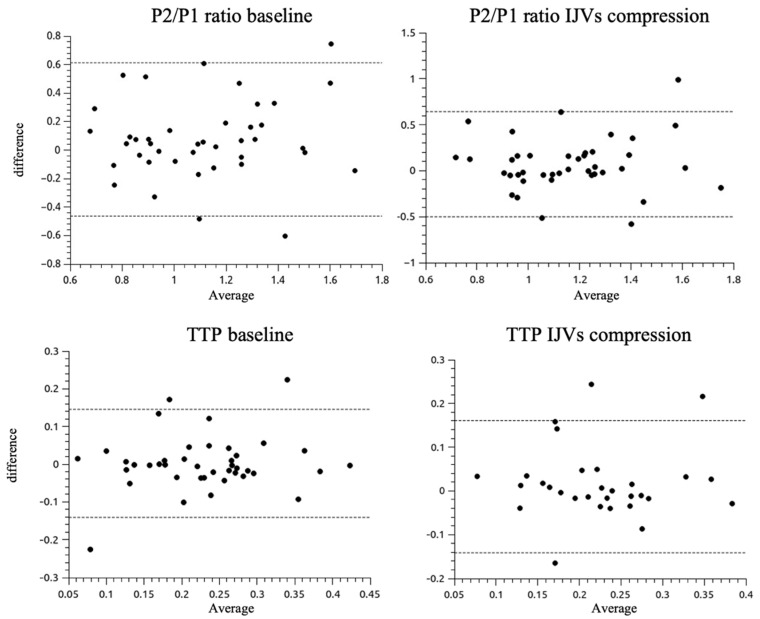
Bland-Altman plots for P2/P1 ratio (up, values from 0.6 up to 1.8) and TTP (down, values from 0.05 up to 0.45) features from ICP and B4C waveform morphologies at the baseline and IJVs compression intervals. The dashed lines represent the 95% limits of agreement (standard deviation 1.96). IJVs: internal jugular veins, TTP: time to peak.

**Figure 5 jpm-11-01302-f005:**
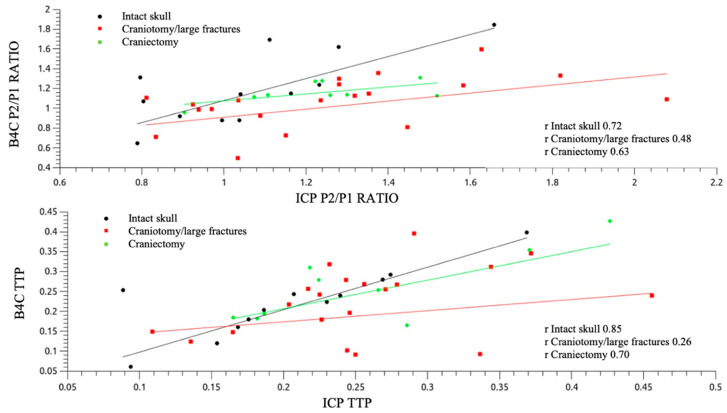
Pearson’s correlation analysis for P2/P1 ratio (**up**) and TTP (**down**) separated between groups according to cranial status.

**Figure 6 jpm-11-01302-f006:**
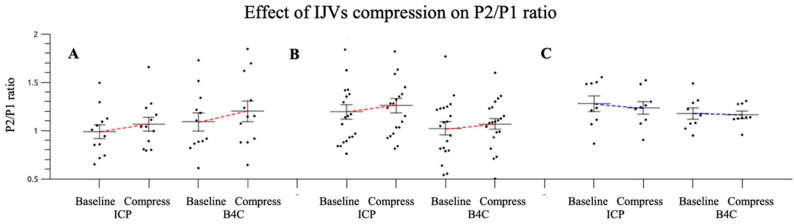
Dispersion plot for both techniques (ICP and B4C). P2/P1 ratio variation according to groups distribution: no cranial damage (**A**, *n* = 12), craniotomy/large cranial fractures (**B**, *n* = 20) and craniectomy (**C**, *n* = 9), regarding ICP elevation intervention. ICP: intracranial pressure, IJVs: internal jugular veins.

**Figure 7 jpm-11-01302-f007:**
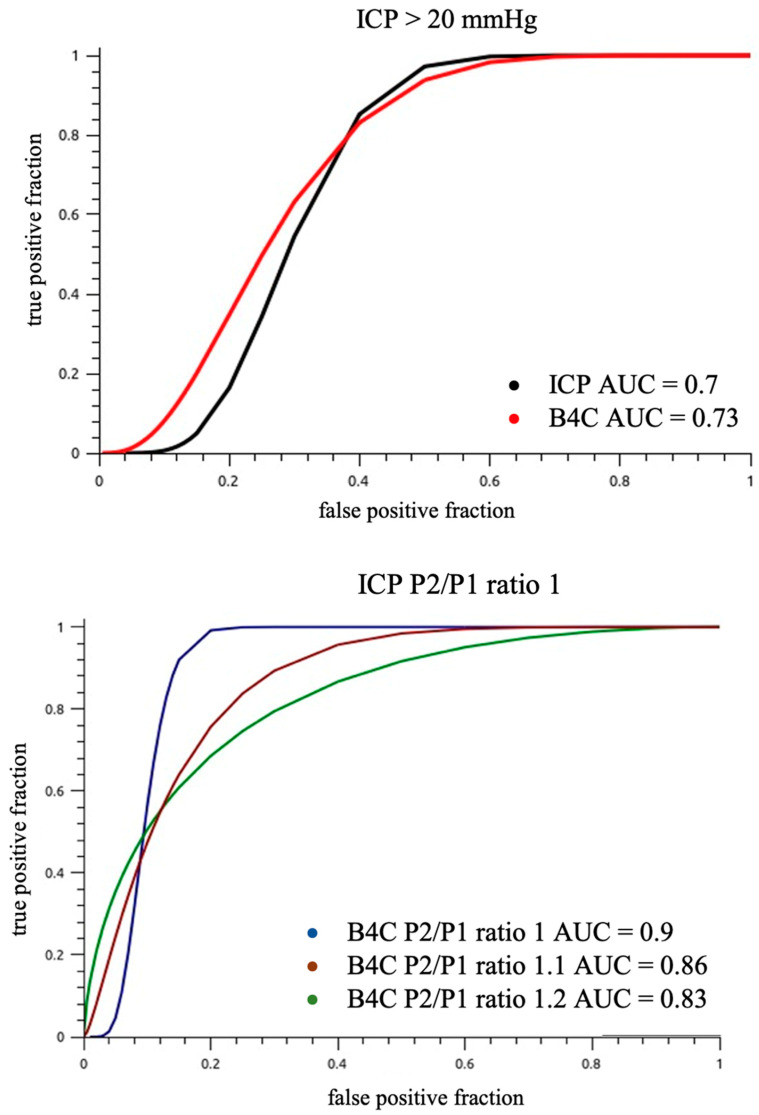
Entire sample ROC analysis of ICP and B4C P2/P1 ratios for an ICP value cut-off >20 mmHg (**up**). The power of predicting intracranial hypertension was reduced because of sample heterogeneity. The B4C P2/P1 ratio cut-off 1 (**down**) was equivalent to ICP P2/P1 ratio cut-off 1 (AUC 0.9).

**Table 1 jpm-11-01302-t001:** Continuous variables are presented as mean ± standard deviation (min; max). Categorical variables presented as n (%). SAPS3: simplified acute physiologic score 3, ICP: intracranial pressure, GCS: Glasgow coma score.

Variable	Total (41)
Age	37.6 ± 28.2 (18; 78)
Male sex	22 (53%)
Pathology	
Traumatic brain injury	21 (51%)
Marshall III	5 (24%)
Marshall V	16 (76%)
Subarachnoid hemorrhage	13 (31%)
Modified Fisher IV	13 (100%)
Stroke	6 (14%)
Tumor	1 (2%)
Neurosurgery	
No	12 (29%)
Craniotomy	20 (48%)
Craniectomy	9 (21%)
Mean arterial pressure	131.4 ± 25.3 (92; 176)
Sedated regimen	
No sedation	5 (12%)
Fentanyl	2 (4%)
Propofol/Fentanyl	16 (39%)
Propofol/Midazolam/Fentanyl	12 (29%)
Thiopental/Fentanyl	6 (14%)
SAPS3	52.5 ± 13.1
Admission GCS	7.5 ± 5.3
Mortality	15 (36%)
ICP baseline	13.83 ± 9.7
ICP compression	17.1 ± 8.2
ICP >20 mmHg during monitoring	15 (36%)

**Table 2 jpm-11-01302-t002:** ICP and P2/P1 ratio variations with IJVs compression. Mean P2/P1 ratio elevation or decrease are presented, as percentages. Data presented as mean ± standard deviation and %. ICP: intracranial pressure, IJVs: internal jugular veins compression.

		Baseline	IJVS Compression
ICP (mmHg)	Intact skull	15.2 ± 7.1	19.3 ± 7.7
	Craniotomy/fractures	15.6 ± 7.4	19.7 ± 7.4
	Craniectomy	20.8 ± 9.4	23.93 ± 8.8
		Δ	%
P2/P1 ratio variation	ICP intact	+0.08 ± 0.5	+8.43
	B4C intact	+0.11 ± 0.6	+10.11
	ICP craniotomy/fractures	+0.07 ± 0.3	+5.60
	B4C craniotomy/fractures	+0.05 ± 0.4	+4.45
	ICP craniectomy	−0.04 ± 0.2	−3.42
	B4C craniectomy	−0.01 ± 0.2	−1.1

## Data Availability

All data from this study is privately archived in the University of São Paulo School of Medicine datasets and may be available as requested.

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
