# Peer review of "A Novel Noninvasive Technique for Intracranial Pressure Waveform Monitoring in Critical Care"

_jpm, 2021, doi:10.3390/jpm11121302_

Round 1
Reviewer 1 Report
In this study the author raises the question whether there is a good correlation between the invasive and non-invasive, respectively, ICP measurements, in a mixed population of patients, having increased ICP.
The authors conclude:
The non-invasive technique provided biometric am-plitude ratios correlated with ICPW variation morphology and is useful for noninvasive critical care monitoring.
Basically, the monitorsystem is well described and meets some important criteria in this topic, since a ideal non-invasive monitor to track ICP should be should be easy to use, accurate, reliable, reproducible, inexpensive and should be associated with minimal infections and haemorrhagic complications.
The study is well written, interesting, and provide a fair suggestion, how to use a non-invasive technique when patients are suspected to have increased ICP, validated against the gold standard, although skull elasticity has been described as a non-invasive technique in animal studies, since 1985. However, it could be more emphasised, whether the authors think the non-invasive technique is suitable, either as as screening tool, rather than technique useful in patients in need of (longer) continuous ICP monitoring?
More specifically
It should be discussed more consistent whether the rather short observation period provide valid results, in this context implicating a longer observation periode could be used to validate the non-invasive technique as a long-term observation possibility, t¨rtaher than a screening tool, used for ICP “snap-shots"
Seen from an anesthetic/ICU point of view it could be intriguing discussing, whether the ICP effect of changes in anesthesthic depth, use of e.g. hypertonic saline/manitol etc could be documented when the described non-invasive technique were used?
The latter issue, might also raise an important question in the context of point-of-care diagnostic tools, implicating the use of non-invasive techniques, when out of hospital patients are suspected of increased ICP, e.g. in the prehospital setting?
Author Response
The study is well written, interesting, and provide a fair suggestion, how to use a non-invasive technique when patients are suspected to have increased ICP, validated against the gold standard, although skull elasticity has been described as a non-invasive technique in animal studies, since 1985. However, it could be more emphasised, whether the authors think the non-invasive technique is suitable, either as as screening tool, rather than technique useful in patients in need of (longer) continuous ICP monitoring?
The section B4C system attributes was improved to clarify this question: “For validation purposes, the present study was short session designed to avoid including confounding factors, such as ABP large variations per example, and focusing on an exclusive fashion to vary ICP. The correlations then were made beat-by-beat between ICP and B4C pulses. However, the system is suitable to continuous long-term monitoring or multiple serials monitoring sessions according to physicians’ judgments, either aiding screening for invasive ICP monitoring per example, but also allowing the observation of ABP, sedation and ventilation changes. Furthermore, the waveform morphologies could reflect the residual compensatory capacity of the brain (Lundberg waves) [1], for subjects with impaired cerebrovascular autoregulation and lower ability for vasodilation and contraction [2]. The aforementioned gives to the ICP waveform a remarkable role in the care of critical patients, since changes in the ICP wave shape may be informative for an incoming or established alteration of the intracranial system [3, 4]. Further studies will estimate the impact of this information on this population’s outcomes.”
More specifically
It should be discussed more consistent whether the rather short observation period provide valid results, in this context implicating a longer observation period could be used to validate the non-invasive technique as a long-term observation possibility, rather than a screening tool, used for ICP “snap-shots"
I think the above paragraph explains well this question, since the system allows continuous monitoring and the short session methodology applied here was exclusively for validation purposes, since we analyzed beat-by-beat (thousands of beats) correlations between waveforms.
Seen from an anesthetic/ICU point of view it could be intriguing discussing, whether the ICP effect of changes in anesthesthic depth, use of e.g. hypertonic saline/manitol etc could be documented when the described non-invasive technique were used?
Personally, I have already performed CSF drainage, ABP variations and ventilation modifications, seeing that the waveform disclosed is very sensitive to these changes, although I have no statistical data for this. In the case of mannitol/saline, a drop for the p2/p1 ratio and TTP would be expected and can be explored in a next study.
The latter issue, might also raise an important question in the context of point-of-care diagnostic tools, implicating the use of non-invasive techniques, when out of hospital patients are suspected of increased ICP, e.g. in the prehospital setting?
As this system is portable and very practical, it is suitable for using anywhere (pre-hospital, ER, outpatient practice, hospitalized and ICU). The main limitation remains for agitated patients that move themselves too much. Making the pin to not maintain a good contact with the skull.

Reviewer 2 Report
The manuscript is a well written presentation of the validation study of B4C in comparison with intracranial pressure wave morphology conducted by th authors.The text presents in a clear , well -structured manner the concept and behind the study design and the methodology used ;as well as the gap that the new technology aims to fill. Good methodology and results section with detailed presentation of the material , statistical analysis and results.Figures and graphs relevant to the study are presented in an overall good way: minor revision needs Figure 4. In the upper Bland-Altman plots for P2/P1 ratio the x axis are in different scales; yet the figures are in the same size and may confuse the reader. I would suggest using the same axis scale for better displaying. Figure 6 would also need to be in larger size for better displaying.Discussion section is well presented with good limitation and future perspective. Relevant and updated reference list.
Author Response
The manuscript is a well written presentation of the validation study of B4C in comparison with intracranial pressure wave morphology conducted by th authors.The text presents in a clear , well -structured manner the concept and behind the study design and the methodology used ;as well as the gap that the new technology aims to fill. Good methodology and results section with detailed presentation of the material , statistical analysis and results.Figures and graphs relevant to the study are presented in an overall good way: minor revision needs Figure 4. In the upper Bland-Altman plots for P2/P1 ratio the x axis are in different scales; yet the figures are in the same size and may confuse the reader. I would suggest using the same axis scale for better displaying. Figure 6 would also need to be in larger size for better displaying.Discussion section is well presented with good limitation and future perspective. Relevant and updated reference list.
Figure 2 was substituted for another with better quality and more explicative. Regarding figure 4, the P2/P1 ratio, as a ratio of P2 amplitude divided by P1 amplitude, gives values ranging around 1, whereas TTP is a measure of time interval, giving values around 0.15 ms, hence, is not possible to put them in the same scale, but I included the ranges displayed in the figure 4 label to reduce the possibility of confusion. Figure 6 was amplified and will be depicted with good quality in the final version.
